# Modelling of the Face-Milling Process by Toroidal Cutter

**DOI:** 10.3390/ma16072829

**Published:** 2023-04-02

**Authors:** Marcin Płodzień, Łukasz Żyłka, Antun Stoić

**Affiliations:** 1Department of Manufacturing Techniques and Automation, The Faculty of Mechanical Engineering and Aeronautics, Rzeszow University of Technology, ul. W. Pola 2, 35-959 Rzeszow, Poland; plodzien@prz.edu.pl; 2Mechanical Engineering Faculty, University of Slavonski Brod, Trg Ivane Brlic Mazuranic 2, 35000 Slavonski Brod, Croatia; antun.stoic.3@gmail.com

**Keywords:** toroidal cutter, cutting force, surface roughness, roughness model, force model

## Abstract

When face milling using a toroidal cutter, with a change in the depth of the cut, the entering angle varies as well. An experimental test of the influence of cutting parameters, such as the depth of the cut and the feed per tooth on the cutting force components and surface roughness parameters, was conducted. The experimental test was carried out using a DMU 100 monoBLOCK CNC machine with registration of cutting force components and surface roughness parameters Ra, Rz, and RSm. FEM analysis of the face-milling process was also carried out and compared with the experimental results. The average deviation of the FEM values for cutting force components does not exceed 12%. Experimental models were established for each force component. It was shown that the depth of the cut has the strongest influence on each force component. The feed per tooth has a little impact on the cutting force. The obtained model of the feed force component is the most complex, and the model of the F_a_ component is only linear. The influence of the *a_p_* parameter on the surface roughness parameters is nonlinear and nonmonotonic. In the range of approx. *a_p_* = 2 mm, there is a maximum surface roughness.

## 1. Introduction

One type of milling process is face milling. This kind of milling process is used mainly in the five-axis milling of sculptured surfaces and also in the milling of flat surfaces [1,2,3]. The face-milling process can be carried out with the use of different types of milling cutters with either a straight or curved cutting edge [4]. Face mills are characterised by the fact that the cutting edge has an entry angle below 90° and is located on the face part of the tool [5].

One of the types of face-milling cutters is the toroidal face cutter [6,7]. Its characteristic feature is having round inserts arranged around the circumference, and the radius of the insert is lower than the radius of the tool. This kind of tool has no nominal entering angle because this angle depends on the cutting parameters (Figure 1). The entering angle is one of the basic geometric parameters of the cutting tool and has a great influence on the cutting process. It has an impact, among others, on the components of the cutting force, the stability of the process, or the topography of the surface [4,8]. The effect of the entering angle on the cutting process was tested by Amigo et al. [9], but only in the turning process. It was shown that the use of low entering angles has a positive effect on tool life. No relationship was found between the entering angle and surface roughness.

In the case of the toroidal cutter, the entering angle changes with the change in the axial infeed value. The higher the value of the axial feed, the longer the cutting edge involved in the cutting process. Therefore, the entering angle increases with increasing axial infeed, which can be seen in Figure 1. This causes the influence of the entering angle *κ_r_* on the cutting process, in the case of the toroidal cutter, to be more complex than for other milling tools.

The toroidal milling process has been the subject of scientific research for many years. Most studies of the toroidal milling process have focused on the five-axis milling process of complex surfaces [10,11,12,13,14,15]. The influence of tool positioning relative to the machined surface was mainly studied. Gdula et al. studied the five-axis milling process of complex surfaces with a toroidal cutter [5,10,16]. The components of the cutting force and the roughness of the surface were shown to depend not only on the tool parameters, the cutting parameters, and the positioning of the tool, but also on the curvature of the machined surface. Zhou conducted a study of the five-axis milling process with a toroidal milling cutter [17]. He developed an analytical model to predict cutting force components dedicated to five-axis milling. The model error was as high as 25%. A lot of research work has been conducted on the basis of the mechanistic force model, but only for ball-end mills. Engin and Altintas [18] developed a model of rectangular and convex triangular inserted cutters. This model can be used to predict cutting forces, vibrations, dimensional surface finish, and stability lobes in milling. The developed cutting force model depends on the chip cross section, tooth contact length, and experimentally determined cutting coefficients. Altintas also developed a universal model to predict the mechanics and dynamics of all cutting operations [19]. This model is based on one general equation and allows the simultaneous prediction of cutting forces, vibrations, and chatter stability. Gonzalo et al., in his work [20], developed a method which allows the determination of cutting coefficients using FEM, with an error of only 4%. The method was not tested for toroidal milling. Lamikiz et al. [21] proposed a method to estimate the cutting force coefficients used in a mechanistic model. They proved that using linear polynomial shear cutting coefficients and constant ploughing cutting coefficients can evaluate cutting forces in the ball-end milling process with an error of less than 15%. They did not verify the developed method for toroidal cutters. The same authors, in their article [22], developed a model for cutting force estimation in sculptured surface milling. They included in the model the effect of part inclination and machining direction. In most of the cases, results with errors below 10% have been obtained. The developed model was tested only for ball-end mills. Merdol and Altintas, in their work [23], performed a simulation of the milling process and optimization strategies to predict and improve the performance of three-axis milling operations with the use of a ball-end mill. The cutting force distribution along the edge-to-part engagement surface was evaluated based on mechanics laws. Urbikain modelled static and dynamic milling forces in inclined operations with circle-segment-end mills [24]. A developed time domain model was used to investigate cutting forces and vibration patterns. Tuysuz, on the other hand, developed a model of the five-axis milling process using a ball-end mill based on a mechanistic model [25]. He proved that modelling the five-axis milling process requires including in the model feed variation contributed by three linear and two rotary motions of the machine tool. Wojciechowski et al. studied edge forces generated in the ball-end milling of inclined surfaces [26]. The highest values of cutting forces were found while milling flat surfaces because of low cutting speed values near the tip of the tool. Therefore, the developed method cannot be used for milling with torus-end mills because a toroidal mill has a different velocity distribution along the cutting edge than a ball-end mill. 

Some of the work involves the study and modelling of surface roughness after milling. Arizmendi and Jiménez developed a model to determine the topography of the surface after face milling [27]. The authors additionally considered tool runout, but mainly analysed surface profiles without roughness parameters. Furthermore, model verification was performed only for selected technological parameters. Similarly, in the publication [28], the authors studied surface roughness after machining with a toroidal cutter. They developed an artificial neural network to predict the roughness parameters Ra and Rt. However, the values of the technological parameters for which the neural network works correctly were not specified. Surface roughness and cutting forces in the face milling of flat surfaces with a toroidal cutter were also studied in [8]. The effects of the number of teeth, the axial length of the inserts, and the angle of tilt on the surface roughness were determined. Statistical analysis of the results was not performed, and the influence of technological parameters was not analysed. Urbikain and de Lacalle developed a new geometrical model that allows the prediction of the surface topography in flank-milling operations with the use of circle-segment-end mills [29]. This model includes almost all the cutting parameters that have an influence on the process: tool geometry, runout, feed rate, and five-axis orientation angles. It was tested in the milling process of a wall made of aluminium alloy, and the tool axis was close to parallel to the workpiece surface; therefore, this model cannot be used in face-milling operations. Arizmendi et al. have also participated in modelling the surface topography in the milling process, but only with the use of ball-end mills [30]. 

An important issue is to ensure the stability of the milling process, especially when machining thin-walled parts or when using tools with long overhang. The stability of the milling process affects, among other things, surface roughness and tool wear. Campa et al. studied the milling process of thin-walled parts with respect to machining stability [31]. A simplified cutting force model was developed and used to determine the stability limit. In turn, in their works [32,33], Campa et al. studied the dynamics of milling thin-walled parts. In their study, they considered the positions of the tool relative to the workpiece as a third variable. They proved that different models have to be used for thin walls and thin floors. Herranz et al. [34] also studied the stability of the low-rigidity milling process elements. They developed a mechanistic simulation model. de Lacalle et al. studied tool deflection in the milling of inclined surfaces [35]. They measured dimensional errors resulting from tool deflection in the high-speed milling of hardened steel surfaces. The main conclusion was that reducing the deflection can be accomplished by judiciously selecting the tool and milling strategy. Salgado et al. studied the influence of various factors of the milling process on the deflection of end mills under cutting forces [36]. Experimental tests proved that the stiffness of the machine and the clamping of the tool are of similar importance in the displacement of the tool tip to the deflection of the tool itself. A torus cutter positioning method was developed to balance the transverse cutting force [37]. A new methodology was developed for the inclusion of vibration analysis and the consideration of energy efficiency in the design of face-milling processes. Furthermore, Lajmert et al. studied the stability of the milling process [38,39]. They used various methods to identify instability during the real machining process based on the Hilbert and Hilbert–Huang vibration decomposition methods. In [40], the paper investigates the stability and dynamic behaviour of the toroidal cutter in five-axis climb milling. The influence of the machining parameters on the stability of only climb milling was highlighted. In [26], the vibration and cutting force were studied in the titanium alloy milling process with a toroidal cutter. A model was developed to determine the waveforms of the cutting force components, taking vibration into account; however, the developed model was tested only for variable feed rates, without taking into account axial feed. In addition, Wagner and Duc studied the toroidal milling process of the Ti-1023 material [7]. The research was mainly focused on cutting tool wear mechanisms, and the results obtained cannot be considered universal.

Niesony, Grzesik, and Habrat studied the nonorthogonal face-milling process of Ti6Al4V titanium alloy with different feed rates [41]. They performed FEM simulations of the cutting process. The simulation results were compared with the experimental data obtained for a similar milling process configuration. In addition, Segonds et al. modelled the milling process with the use of a toroidal cutter [42]. They developed an analytical model that takes into account the effect of the feed rate for the calculation of the scallop height.

Most of the research is based on a mechanistic model of the components of the cutting force and focusses on determining the cutting coefficients. In addition, most of the research is concerned with cylindrical and ball-end mills. There are no experimental models available of the components of the cutting force in direct relation to the cutting parameters. Furthermore, there is a lack of analysis and experimental verification for horizontal milling with a toroidal cutter. In addition, from the work in [24], the results showed that a small change in the geometry of the tool makes it necessary to develop a customised model; hence, it is necessary to study toroidal cutters due to the lack of available models. In addition, models developed for ball-end mills cannot be used to model milling with toroidal cutters because of the different distribution of cutting speed along the cutting edge, among other reasons.

First, FEM analyses were performed to determine the effects of the cutting parameters on the components of the cutting force. The cutting conditions recommended by the tool manufacturer were adopted. Next, an experimental study was conducted to verify the FEM analyses and experimentally confirm the observed relationships. The aim was to confirm the suitability of the power law model implemented in AdvantEdge 7.9, Third Wave Systems, Minneapolis, MN, USA software for modelling face milling with a toroidal cutter. The experimental results obtained were statistically analysed, and experimental models of the cutting force components were developed. Then, the analysis of selected surface topography parameters was carried out. The preliminary analysis indicated the need to repeat the experiment with the use of coolant to avoid adhesion. Again, the surface was analysed after machining with coolant, and experimental models were determined for three roughness parameters, Ra, Rz, and RSm.

## 2. Numerical and Experimental Procedure

### 2.1. Numerical Background

The first stage of the work was to carry out simulation studies of the face-milling process with a toroidal mill using the FEM method. Simulation studies were carried out using the AdvantEdge software. To precisely represent the cutting tool and reproduce the geometry of the edge, a scan of a part of the cutting insert was performed (Figure 2a). A cutting insert model was then created from the scan and implemented in AdvantEdge. (Figure 2b). 

Subsequently, a workpiece material was defined, whose properties corresponded to 42CrMo4 steel. Numerical analyses were carried out on the basis of a constitutive model of the material in the form of a power law function:(1)$$\sigma (\varepsilon^{p},\dot{\varepsilon },T)=g(\varepsilon^{p})\cdot \Gamma (\dot{\varepsilon })\cdot \Theta (T)$$
where $g(\varepsilon^{p})$ is strain hardening, $\Gamma (\dot{\varepsilon })$ is strain rate sensitivity, Θ(T) is thermal softening, $\varepsilon^{p}$ is plastic strain, $\dot{\varepsilon }$ is strain rate, and *T* is temperature. The strain hardening function can be expressed as:(2)g=σ01+εpε0p1n, if εp<εcutp
(3)g=σ01+εcutpε0p1n,if εp≥εcutp
where $\sigma_{0}$ is initial yield stress, $\varepsilon^{p}$ is plastic strain, $\varepsilon_{0}^{p}$ is reference plastic strain, 1n is the strain hardening exponent, and $\varepsilon_{cut}^{p}$ is the strain hardening cutoff. The equation of strain rate sensitivity can be written as:(4)Γ(ε˙p)=1+ε˙pε˙0p1m1, if ε˙p≤ε˙tp
(5)Γε˙p=1+ε˙pε˙0p1m21+ε˙tpε˙0p1m1−1m2,if ε˙p>ε˙tp
where ${\dot{\varepsilon }}^{p}$ is plastic strain rate, ${\dot{\varepsilon }}_{0}^{p}$ is reference plastic strain rate, ${\dot{\varepsilon }}_{t}^{p}$ is threshold plastic strain rate, *m*_1_ is the low strain rate sensitivity exponent, and *m*_2_ is the high strain rate sensitivity exponent. The thermal softening function is expressed as
(6)$$\Theta \left(T\right)=c_{0}+c_{1}T+c_{2}T^{2}+c_{3}T^{3}+c_{4}T^{4}+c_{5}T^{5},\;\mathrm{if}\;T<T_{cut}$$
(7)$$\Theta \left(T\right)=\Theta \left(T_{cut}\right)-\frac{\left(T-T_{cut}\right)}{\left(T_{melt}-T_{cut}\right)},if\;T\geq T_{cut}$$
where *c*_0_, *c*_1_, *c*_2_*, c*_3_, *c*_4_, and *c*_5_ are coefficients of the fifth-degree polynomial, *T* is temperature, *T_cut_* is cutoff temperature, and *T_melt_* is temperature of melting. 

Simulation studies were carried out for the kinematics of the milling process shown in Figure 3a. The tool performs a rotary motion with a cutting speed *v_c_* and a feed motion with a feed rate *v_f_*. The tool feed is described by two parameters, the axial feed *a_p_* and the radial feed *a_e_*. A finite element mesh was generated for the system thus adopted (Figure 3b). Only the motion of one cutting insert (one blade) was simulated in order to reduce the finite elements and shorten the calculation time. The surface of the cutting edge’s radius of roundness was compacted with elements equal to 1/3rd of the blade’s radius of roundness in order to increase the accuracy of the calculations and to establish the correct contact conditions between the blade and the material. A coulomb friction model was used, which was built into the AdvantEdge software. Based on [43], the coefficient of friction for coating a tool-steel pair was set as 0.4. An example of visualisation of the FEM analysis during chip formation is shown in Figure 4.

### 2.2. Experimental Procedure

After simulation studies, experimental research was carried out. They were caried out using the DMU 100 monoBLOCK CNC machine. The milling machine was equipped with a plate piezoelectric dynamometer Kistler 9257B with a charge amplifier module Kistler 5070 and a digital analogue converter NI USB-6003 from National Instruments. The view of the workspace is shown in Figure 5. Signal processing was performed using NI SignalExpress 2013 V7.0, Austin, TX, USA software. Measurements of surface roughness after the milling process were made using the Mahr MarSurf 300 profilometer. Surface roughness was measured at five locations and then the average value of roughness was determined. A dispersion of less than 10% was obtained in all measurements. The measurement was made along the feed direction in the centre of the cutting width. The elementary length was equal to 0.8 mm and the measurement length was equal to 5.6 mm, according to PN-EN ISO 21920:2022.

The milling process tests were carried out on 42CrMo4 steel with a hardness of 220 HB. This is a steel that is commonly used primarily for highly stressed parts of medium and greater thicknesses, with high strength requirements. The main applications include axles, shafts, crankshafts, connecting rods, machine tool spindles, gears, sprockets, cylinders, ball pins, multispline shafts, and other machine parts subjected to varying bending and torsional loads and wear. The chemical composition of 42CrMo4 steel is listed in Table 1.

The tests were carried out using a Sandvik Coromant R300-050Q22-08H face-milling cutter, 42 mm in diameter, with one R300-0828E-PL 1030 circular insert with a radius of 4 mm dedicated to steel finishing mounted. The insert was made of cemented carbide and coated with PVD TiAlN coating. The rake angle was equal to 7° and the relief angle was 8°. The milling cutter had only one cutting insert mounted to replicate the cutting conditions as in the simulation tests. This arrangement provided a clear reading of the cutting force and also avoided the influence of axial run-out of the blades on the test results. A new cutting insert was used for each cutting test to eliminate the influence of blade wear on the test results.

The experimental investigations were planned according to the statistical plan, a central composite fractional design with 3 levels and 2 factors. 42CrMo4 steel samples were face-milled with varying axial depths of the cut *a_p_* and feeds per tooth *f_z_*, obtaining different values for the entering angle. The test plan is shown in Table 2. 

According to the test plan, the selected tests were repeated to eliminate the influence of other factors on the test results. The machining was carried out according to the tool manufacturer’s recommendations, without the use of coolant. Due to the strong appearance of buildup on the machined surface, cutting tests were repeated using a coolant: Fuchs Ecocool 68 CF semisynthetic emulsion with a concentration of 6%. Cutting parameters dedicated to the cutting insert were used. Milling was carried out concurrently with a constant cutting speed (*v_c_* = 230 m/min) and a constant radial infeed (*a_e_* = 20 mm). The cutting force and surface roughness after machining were measured in each cutting test. The components of the cutting force were measured in the workpiece coordinate system, as shown in Figure 5.

## 3. Results and Discussion

### 3.1. Cutting Force Components Analysis

The basic signal that characterises the cutting process is the cutting force. Therefore, the components of the cutting force were determined for the machining conditions specified in Table 2. The cutting forces were first determined in AdvantEdge using the FEM method for the adopted material model. Experimental tests were then carried out for the same cutting conditions, and the values obtained were compared. In each test, the feed component F_f_, the component normal to the feed direction F_fN_, and the axial component F_a_ of the cutting force were recorded and determined. The obtained values of the cutting force components are shown in Table 3.

A comparison was made between the measured and numerically calculated values. The analysis shows that the FEM calculations are very close to the actual values. In the case of the F_fN_ component, the average deviation of the FEM values was approximately −9%. For the F_f_ component, it was the largest at −12%, and for the axial component Fa, the deviation was the smallest at only 4%. In the case of the F_f_ component, the largest differences between actual and simulated values occurred for test repetitions (f_z_ = 0.1, a_p_ = 1.5) and were equal to approximately −20%. For the F_f_ and F_fN_ components, the FEM-determined values were lower than the actual cutting force values. On the contrary, for the axial component, the simulated values were higher than the actual forces in most of the cutting tests. However, using FEM analysis, results with values very close to the measured cutting force were obtained. This means that the material model and the values of the model coefficients were assumed correctly. In the case of FEM modelling of the milling process with a ball-end mill, different relationships were obtained between the real data and the simulation values. A slight overestimation in the main cutting force and in the axial force was observed, while values below the experimental forces are observed in the radial force [20]. Therefore, it is not possible to use ball-end mill models to model the face-milling process with a toroidal cutter.

Then, an analysis of the variability of the cutting force components was carried out, depending on the values of the changed technological parameters. Particular attention was paid to the depth of the cut, as it influences the entering angle. The actual values of the cutting force components recorded during the cutting tests were analysed. Figure 6a–c shows the variation in the cutting force components as a function of the depth of the cut, and Figure 6d–f shows them as a function of the feed. The graphs show that the influence of the two technological parameters on the cutting force components varies strongly. This is probably due to two facts. Both technological parameters mainly affect the cross-sectional area of the cut layer. Additionally, the feed rate influences the actual thickness of the chip and the specific cutting resistance, while the depth of the cut affects the actual entering angle. Thus, in face milling with a toroidal cutter, these relationships overlap.

Analysing the graphs of the variation in force components, it can be observed that the F_fN_ force has the highest value. In most cases, its value is about twice that of the other force components. The components F_f_ and F_a_ have similar values, and their functions vary depending on the technological parameters.

For constant values of feed per tooth (Figure 6a–c), the change in all components of the cutting force has a linear form. However, it can be seen that depending on the value of feed per tooth, the intensity of the increase in the components F_f_ and F_a_, and their relationship to each other, changes. For a small value of feed per tooth (*f_z_* = 0.05 mm/tooth), the F_a_ component is clearly greater than the feed component by about 50%. For a feed rate twice as large, the difference decreases. In contrast, for a feed rate of *f_z_* = 0.15 mm/tooth, the F_f_ component is slightly larger than the axial component. The effect of feed per tooth on the axial and feed components of the cutting force can be clearly seen in the graphs of Figure 6d–f. It can be observed that, as the feed per tooth increases, the feed component F_f_ increases much more strongly than the axial component. It can even be said that the value of the axial force component hardly depends on the feed per tooth. On the other hand, the F_f_ component increases monotonically as the parameter *f_z_* increases. This increase is so strong that at a feed rate of 0.15 mm/tooth, the F_f_ component exceeds the value of the F_a_ component. This can be explained by the fact that, as the feed rate increases, the cross section of the cut layer increases only in the feed direction, and the shape of the layer does not change. Therefore, the higher the feed rate, the greater the resistance that acts on the tool in the feed direction. This results in an almost proportional increase in the F_f_ component.

Analysing the effect of the depth of the cut on the components of the cutting force, it can be seen that, regardless of the value of the parameter ap and the entering angle, the functions of each component do not change. Force components increase linearly with an increasing depth of cut ap. However, analysing the graphs in Figure 6d–f, one can see the effect of depth of the cut on the values of the components and the relationship between them. For example, for a_p_ = 0.5 mm, the components Fa and Ff were, at most, equal to 60% of the value of the component FfN; for depth a_p_ = 1.5, the difference was already only 45%; and for the highest cut depth, the difference decreased to only 38%. This means that the greater the depth of the cut, the stronger the increase in the F_fN_ component, and the smaller the increase in the other components Fa and Ff. This means that the greater the depth of the cut and, at the same time, the entering angle, the greater the part of the load transferred in the direction normal to the feed, that is, in the direction of the workpiece. This means that for higher entering angles, the load on the workpiece is greater. For small cutting depths, the load in the direction of the workpiece is greater than the load transferred in the axial direction and only twice as great in the direction of the feed. For a_p_ = 2.5 mm, the difference is almost three times.

One more fact can be noted. For small values of the depth of cut, the most advantageous cutting conditions are obtained in terms of the cutting force components. For the smallest feed rate, all components of the cutting force are almost equal to each other. However, for the largest depth of cut tested, the F_fN_ component stabilises and does not increase with an increasing feed rate.

To confirm the validity of the conclusions drawn, a statistical analysis of the experimental results was carried out. Mathematical models were developed for each component of the cutting force with the best fit, and a variance analysis was performed. For each component of the cutting force, an equation was determined, and statistical parameters were determined to evaluate the resulting models. A significance level of α = 0.05 was adopted.

For the normal-to-feed cutting force component F_fN_, the following relationship was obtained:F_fN_ = 71.7 + 323∙*f_z_* + 116.7∙*a_p_* + 1640 *f_z_*∙*a_p_*,(8)
which fits very well with the experimental data, as the coefficient R^2^ = 0.99. A graphical representation of the model is shown in Figure 7. Table 4 shows the results of the analysis of variance of the generated model. 

It can be seen that the response surface is almost flat because there are no square factors in the model. However, the degree of slope of the surface in each direction varies and depends on the values of the technological parameters. It can be seen that the dominant technological parameter is the depth of the cut. It has the strongest effect on the normal-to-feed cutting force component. The contribution of the depth-of-cut factor *a_p_* in the model is as much as 76%, and the F-value is many times greater than that of the other factors. The influence of the depth of the cut is stronger, the greater the feed per tooth. The parameter *f_z_* is also significant in the equation obtained, but its F-value is four times lower than for the *a_p_* parameter. However, as the variance results and the shape of the response surface show, its effect on the F_fN_ force is much smaller. This means that increasing the feed rate does not cause such a significant increase in the F_fN_ force, as in the case of increasing the depth of the cut. In addition, contribution analysis indicates the largest contribution of the *a_p_* parameter to the model. All factors of the model are statistically significant because its *p*-value is close to zero. Moreover, the *p*-value for the lack-of-fit test is close to the level of significance, and the F-value for the lack of fit is many times lower than for model factors, which indicates a small model error. The product of the depth of the cut and the feed rate has the least influence on the normal-to-feed force. Its contribution is the smallest, as confirmed by the contribution and the F-value.

The next force analysed was the feed component of the cutting force F_f_. The obtained model of the feed force differs from the model of the component normal-to-feed because it includes a quadratic factor and is as follows:F_f_ = 101.2 − 1264∙*f_z_* + 29.12∙*a_p_* + 9164 *f_z_*^2^ + 606.6 *f_z_*∙*a_p_.*(9)

The resulting model has one quadratic factor and one two-way interaction factor in addition to the linear factors. This results in a response surface that is not flat and has curvature (Figure 8). Notable is the fact that the square factor is the feed per tooth. This is due to the dependence of the specific cutting resistance on the feed rate. The higher the feed rate, the lower the resistance of the workpiece material. In addition, the higher the feed rate, the greater the cross section of the cut layer. The superimposition of these two phenomena causes the feed component to depend nonlinearly on the feed (Figure 8). 

By analysing the response surface, it can be seen that the depth of the cut has a proportional effect on the force F_f_. Regardless of the value of the feed per tooth, the force F_f_ varies linearly with a change in the depth of the cut. On the contrary, the increase in the F_f_ force caused by the feed per tooth is much smaller. Similar to the F_fN_ force component, feed per tooth has less influence on the force value than on the depth of the cut. The depth of the cut has the largest contribution (more than 55%) to the model and the largest F-value, confirming its dominance (Table 5). Feed per tooth in linear form also has a significant contribution (almost 38%), and its F-value is slightly lower than for the depth of cut. The influence of nonlinear factors is small; their contribution is only about 6%, which is less than a two-way factor. Each factor in the model is highly significant, which is confirmed by the low (close to zero) *p*-value. In addition, the values of the parameter of the F-value confirm the model analysis performed. Lack-of-fit parameters, such as contribution, *p*-value, and F-value, indicate a small error of the model. 

The third component of the cutting force that was analysed was the axial component of F_a_. It acts in the direction of the tool axis, and its effect on the cutting process is of secondary importance. However, it is an advantageous situation to transfer the load in the axial direction of the tool because, in this direction, the tool has the greatest stiffness. A model was created to describe the Fa component, which has the following form:F_a_ = 69.4 + 403∙*f_z_* + 94.67∙*a_p_.*(10)

The Fa axial component model has the simplest form. It contains only linear components. It does not include quadratic components and two-way interaction factors. However, the fit of the model to the experimental data is very good, with a coefficient of R^2^ = 0.97. Such a simple form of the model is due to the fact that there is no motion in the direction of the tool axis and the cutting process, i.e., the cutting speed vector and the feed vector act in a plane perpendicular to the tool axis. As can be seen from the response surface analysis presented in Figure 9 and the results of the analysis of variance presented in Table 6, it is the depth of the cut that has the greatest influence on the axial component of the cutting force. The contribution of the *a_p_* factor in the model is as high as 93%, which means that it is the most important. The feed-per-tooth parameter has a very small F-value compared to *a_p_*. Moreover, the *p*-value parameter for feed per tooth has the highest value. Both facts prove that the feed rate in the obtained model is of little importance. This is because the feed is carried out in the direction perpendicular to the tool axis and axial force component. 

### 3.2. Surface Roughness Analysis

After each milling test, the basic surface roughness parameters were measured and then an attempt was made to fit the mathematical models. However, the developed models showed a very poor fit to the experimental data at R^2^ < 0.5. The surface roughness in some of the milling tests varied independently of the technological parameters. Hence, images of the surfaces after milling were analysed. Observing the surfaces at 50 and 200 times magnification, large amounts of workpiece material attached to the machined surface were observed (Figure 10).

The irregularly distributed particles of workpiece material remaining on the surface are the result of adhesion phenomena. Despite using the cutting parameters and machining conditions recommended by the tool manufacturer, it was not possible to achieve a good surface quality. The adhesion phenomenon occurred regardless of the values of the technological parameters. Therefore, cutting tests were repeated using a semisynthetic water-based coolant emulsion with a concentration of 5% and surface roughness was measured again. A view of the surfaces obtained after milling with coolant is shown in Figure 11. It can be seen that adhesion phenomena were almost completely eliminated. In Figure 11b, it can be clearly seen that traces of cutting insert passes dominate and adhesion of the workpiece material is sporadic.

Three surface roughness parameters, Ra, Rz, and RSm, were analysed. A significance factor of α = 0.05 was assumed. An analysis of variance was used to determine the significance of the influence of the milling process parameters on the surface roughness parameters. The experimental models were then determined, and the model fit was established.

For the roughness parameter Ra, the following relationship was obtained:Ra = 0.0882 − 0.454∙*f_z_* + 0.0842∙*a_p_* + 5.93∙*f_z_*^2^ − 0.02283∙*a_p_*^2^,(11)
which has a very good fit to the experimental data, as the coefficient R^2^ = 0.95. A graphical presentation of the model is shown in Figure 12. In turn, Table 7 shows the results of the analysis of variance for the developed model.

In the case of the Ra parameter, all the factors in the model have a high impact significance, but not the same. The most significant parameter is the feed rate. The depth of cut, in linear and quadratic form, is characterised by slightly lower significance. In turn, the square of the feed rate *f_z_*^2^, which is present in the model, is the least statistically significant, although its *p*-value of 0.023 is greater than the accepted level of significance. This is also confirmed by the very small contribution of this factor to the overall model, at only 1%. The strongest influence on the roughness parameter Ra is the feed rate *f_z_*. The contribution of this factor in the model is almost 70%. This can also be seen in the graphical representation of the model, as the response surface is more curved in the feed direction. However, the dependence of Ra on the feed rate *f_z_* is not linear, since two factors, *f_z_* and *f_z_*^2^, contribute to the model. The dependence of the roughness parameter Ra on the depth of the cut *a_p_* is very interesting. In the model, both the linear and the quadratic factors of the *a_p_* parameter are involved. The significance of the impact of the depth of the cut is lower than that of the feed per tooth, and this impact is noticeable. The influence of the *a_p_* parameter on the Ra parameter is nonmonotonic. It can be observed that, in the range of approximately *a_p_* = 2 mm, there is a maximum Ra function in the entire range of feed variation per tooth. For higher feed rates, the function stabilises, and the curvature in the direction of variation of the *a_p_* parameter is smaller. It can be concluded that the higher the feed rate, the smaller the dependence of the Ra parameter on the depth of the cut. From the obtained model, it can be seen that, to obtain the lowest roughness, it is necessary to mill with the lowest feed rate and a cut depth greater than or less than 2 mm. This means that the highest value of the Ra parameter is obtained at an entering angle of 45° (*a_p_* = 2 mm).

The roughness parameter Rz was also analysed and is described by the following relationship:Rz = −0.455 + 7.8∙*f_z_* + 1.601∙*a_p_ −* 0.503∙*a_p_*^2^,(12)

The resulting model has a slightly poorer fit to the experimental data than to the Ra parameter. However, a very good fit was obtained, with a coefficient of R^2^ = 0.84. A clear difference can be seen between the Ra and Rz parameter models. There is no factor *f_z_*^2^ in the equation of the Rz parameter, indicating only a linear relationship between the Rz parameter and the feed rate. This relationship is confirmed by the graphical interpretation of the model shown in Figure 13. It can be clearly seen that the resulting surface is not curved in the feed direction. To analyse the model in detail, the results of the analysis of variance is included in Table 8. It can be seen that the fit error is less than 16%, which can be considered to be a good result. Moreover, the *p*-value for the lack-of-fit test is much lower than the level of significance, which confirms that the model is correct. However, with the analysis of the contribution of the factors in the model and the F-value, it can be concluded that the *a_p_* factor has the least influence on the value of the Rz parameter and can be classified as insignificant (its *p*-value is lower than α). In contrast, the factor *a_p_*^2^ shows a significant impact on the value of the Rz parameter because the *p*-value = 0.001, as for the other components of the model. This is confirmed by the response surface generated. It is characterised by a parabolic shape in the direction of variation of the parameter *a_p_*. The function of the Rz parameter, irrespective of the feed per tooth, has a maximum at *a_p_* = 1.8 mm. This means that cutting with a depth of cut approximately half the radius of the cutting insert produces the worst surface roughness.

The RSm roughness parameter was also analysed in addition to the Ra and Rz parameters. For the RSm roughness parameter, a relationship was obtained whose form is similar to the model for the Rz parameter:RSm = −28.67 + 330∙*f_z_* + 72.44∙*ap* − 21.33∙*a_p_*^2^,(13)

The obtained relationship is characterised, similarly to the Ra parameter, by a very good fit to the experimental data, as the coefficient R^2^ = 0.96. On the other hand, similarly to the model for the Rz parameter, the *f_z_*^2^ factor is not present. This means that the RSm parameter, like Rz, is characterised by a linear dependence on the feed to the cutting edge, which is proven by the graphical presentation of the model in Figure 14. The function of the RSm parameter is very similar to that of the Rz parameter, but the depth-of-cut parameter is as significant as the others. In the cutting depth direction, the shape of the function is parabolic, with a maximum at *a_p_* = 1.8 mm. However, it can be clearly seen that the value of the RSm parameter for greater depths of cut is significantly higher than for the smallest depth of cut. The results of the analysis of variance for the developed model confirm the analysis. The values of the F-value and *p*-value parameters, which have almost identical values for all factors in the model, are notable (Table 9).

Analysis of the experimental models developed for the roughness parameters indicates a better fit of the Ra and RSm parameter models than Rz. This is because the Rz parameter is sensitive to any surface irregularities that may occur randomly. The Ra and RSm parameters average out the roughness profile. Despite the use of coolant, the adhesion phenomenon, which is characteristic of steel cutting, has not been completely eliminated. Therefore, the Rz parameter has a greater dispersion than the expected value. This is also confirmed by the analysis of variance of the models obtained. The analysis also showed that the worst surface roughness is obtained for depths of cut close to half the cutting insert radius. In this case, the rake angle is 45°.

## 4. Conclusions

Experimental and simulation studies of the face-milling process on a flat surface with the use of a toroidal milling cutter were carried out. The convergence of simulation analyses, using the FEM method with experimental results, was demonstrated. The simulation error for each component of the cutting force was estimated. The largest average error was equal to −12% for the F_f_ component. The average deviation of the FEM values for the F_fN_ component was approximately −9%, and the smallest deviation was for the axial component F_a_, which was equal to only 4%.

An analysis of the effects of technological parameters, feed per tooth and depth of cut, on the cutting force components was carried out. Analysis shows that the depth of the cut has a stronger effect on cutting force components than feed, with the strongest effect on the F_fN_ component. The greater the depth of the cut and entering angle, the greater the part of the load transferred in the direction normal to the feed (in the direction of the workpiece). Furthermore, for small depths of cut, the F_fN_ is two times greater than the F_f_ and F_a_, but for *a_p_* = 2.5 mm, the difference is almost three times. For the smallest feed rate, all components of the cutting force are almost equal to each other. The model of the force component Ff is the most complex because it has linear, square, and two-way indicators, but the model of F_a_ is linear, with the dominant significance of the *a_p_* parameter.

Analysis of the machined surface showed that the face milling of 42CrMo4 steel must be performed with coolant to reduce the effect of chip adhesion to the workpiece surface. The worst surface roughness was obtained for depths of cut close to values equal to half the radius of the cutting insert when the entering angle was close to 45°. This is because the influence of the *a_p_* parameter on the surface roughness parameters is nonlinear and nonmonotonic. In the range of approx. *a_p_* = 2 mm (κ_r_ = 45°), there is a maximum of functions in the entire range of the feed-per-tooth variation. When planning a face-milling process with a toroidal cutter, cutting depths with values close to half the radius of the cutting insert should be avoided. Greater or smaller depths of cut should be used to minimise surface roughness.

Further research should be conducted on the modelling of the toroidal milling process using other cutting-blade geometries, especially negative ones for medium and heavy machining. In addition, the developed models must be verified for negative geometries, high feed rates, and large cut depths.

## Figures and Tables

**Figure 1 materials-16-02829-f001:**
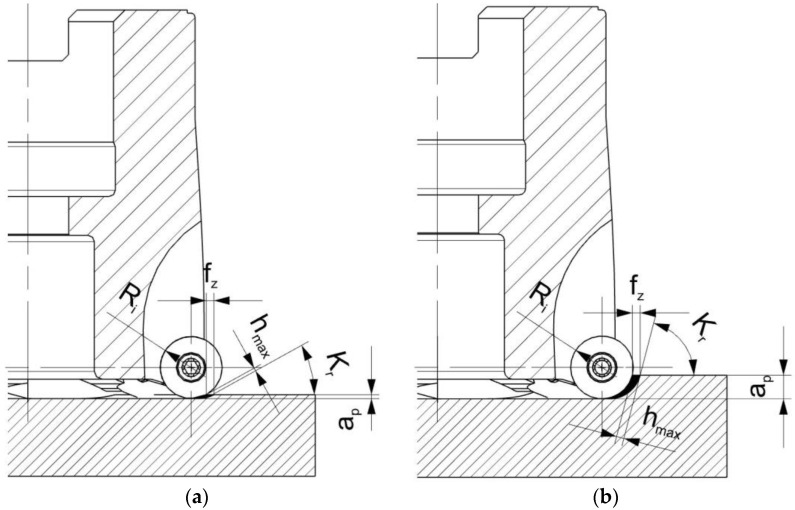
The influence of axial infeed on the entering angle: (**a**) small axial infeed; (**b**) large axial infeed.

**Figure 2 materials-16-02829-f002:**
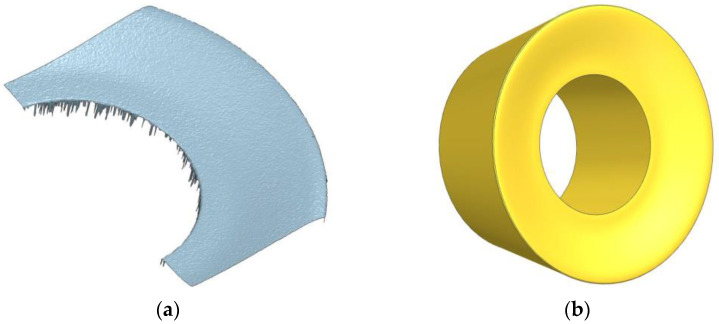
Reconstruction of cutting edge geometry: (**a**) cutting insert scan; (**b**) cutting insert model.

**Figure 3 materials-16-02829-f003:**
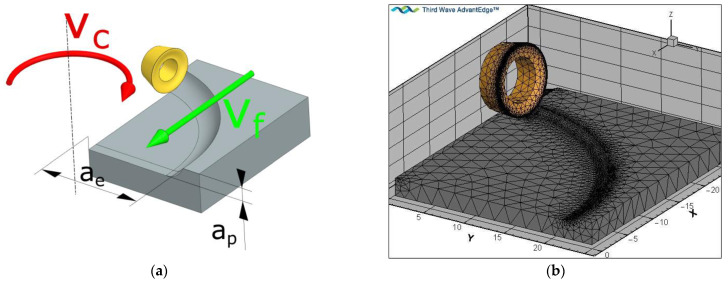
Kinematics of the milling process: (**a**) milling parameters; (**b**) finite element mesh.

**Figure 4 materials-16-02829-f004:**
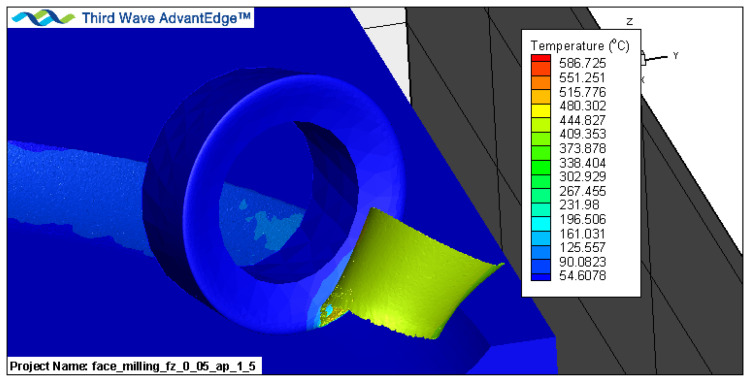
FEM analysis of chip formation.

**Figure 5 materials-16-02829-f005:**
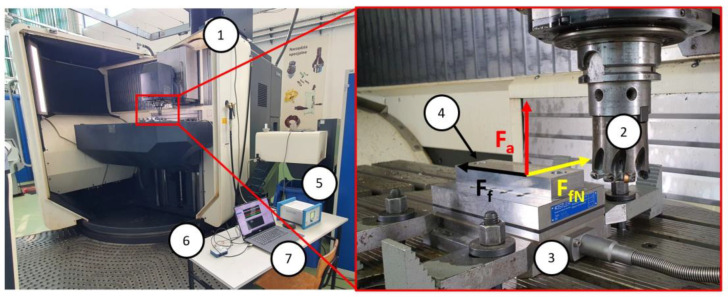
View of the test stand and machine workspace: (1) machine tool; (2) cutting tool; (3) dynamometer; (4) test sample; (5) charge amplifier; (6) A/C converter; (7) computer.

**Figure 6 materials-16-02829-f006:**
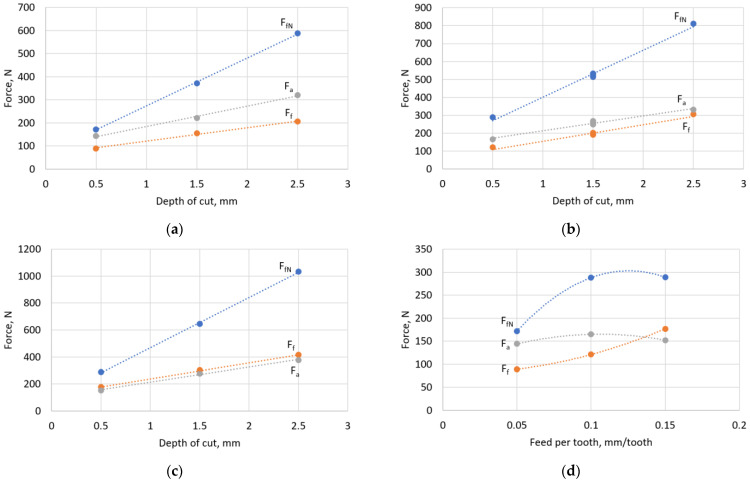
Components of the cutting force: (**a**) in the function of depth of cut, *f_z_* = 0.05; (**b**) in the function of depth of cut, *f_z_* = 0.1; (**c**) in the function of depth of cut, *f_z_* = 0.15; (**d**) in the function of feed, *a_p_* = 0.5; (**e**) in the function of feed, *a_p_* = 1.5; (**f**) in the function of feed, *a_p_* = 2.5.

**Figure 7 materials-16-02829-f007:**
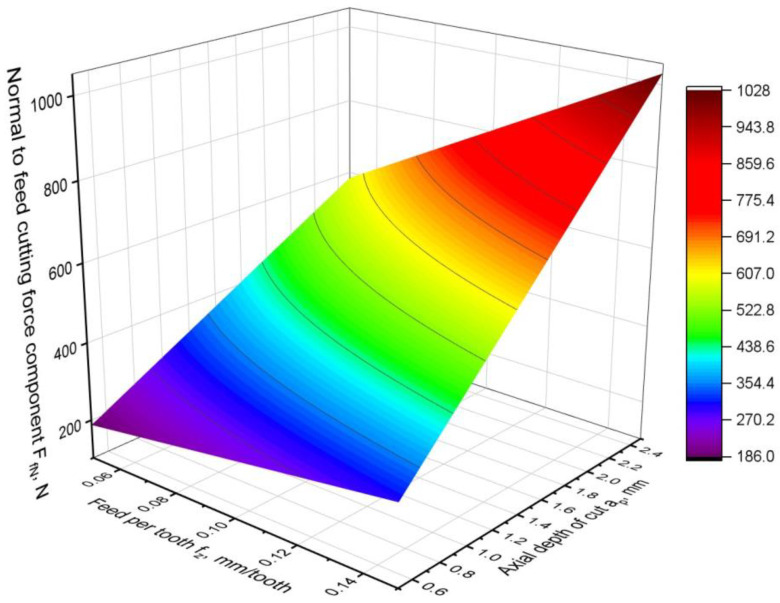
Normal-to-feed cutting force component F_fN_ model.

**Figure 8 materials-16-02829-f008:**
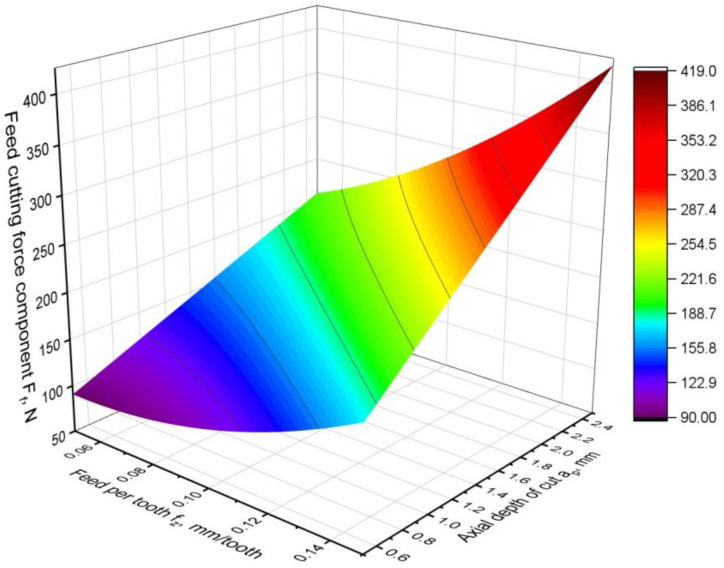
Feed cutting force component F_f_ model.

**Figure 9 materials-16-02829-f009:**
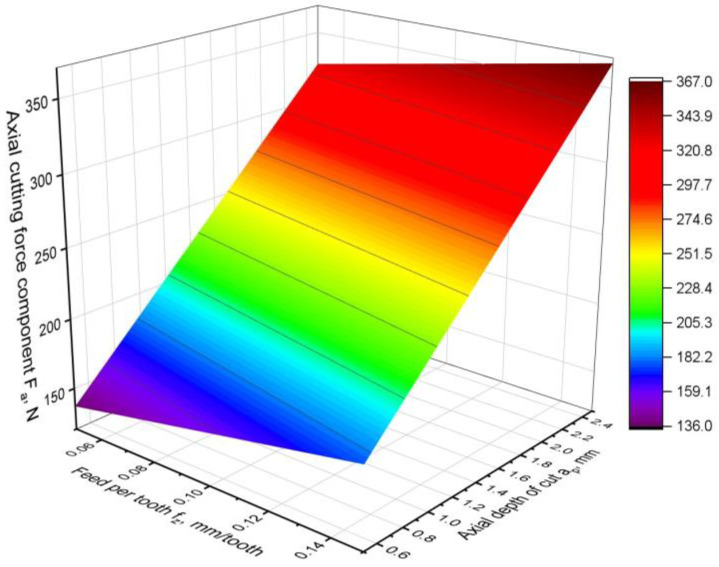
Axial cutting force component F_a_ model.

**Figure 10 materials-16-02829-f010:**
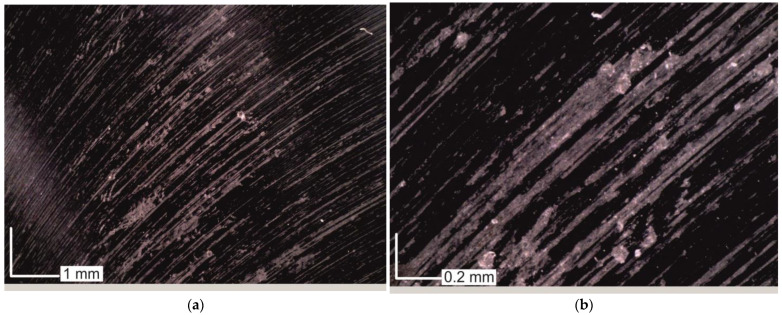
Surface after machining without cooling: (**a**) 100 times magnification; (**b**) 200 times magnification.

**Figure 11 materials-16-02829-f011:**
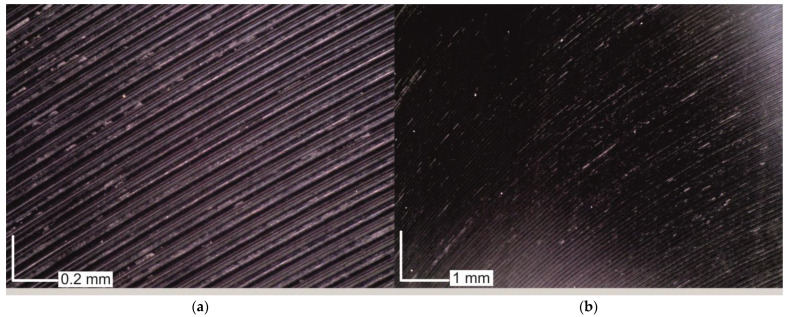
Surface after machining with the use of cooling: (**a**) 100 times magnification; (**b**) 200 times magnification.

**Figure 12 materials-16-02829-f012:**
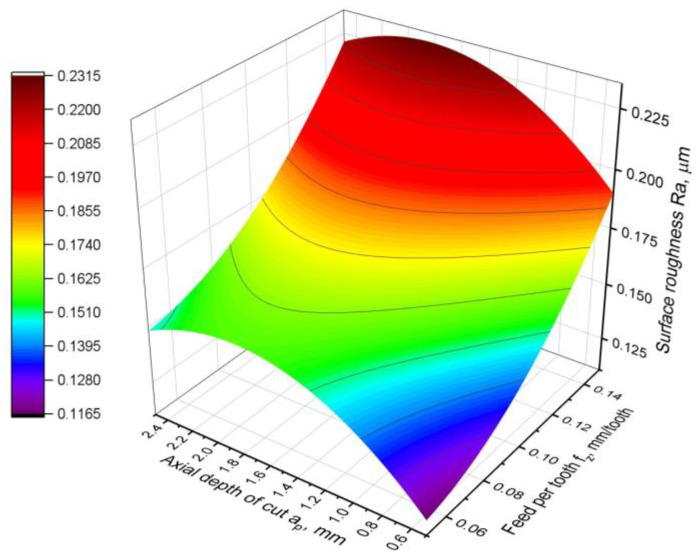
Surface roughness Ra parameter model.

**Figure 13 materials-16-02829-f013:**
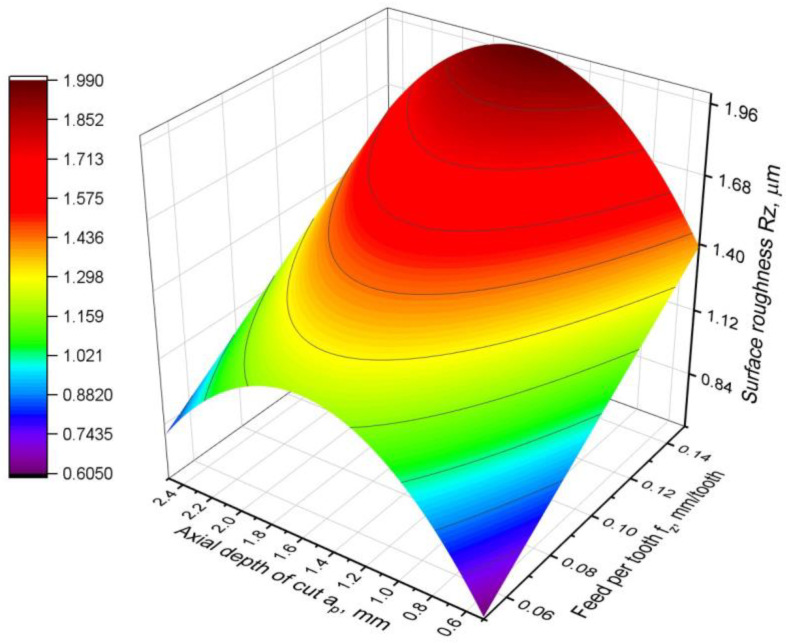
Surface roughness Rz parameter model.

**Figure 14 materials-16-02829-f014:**
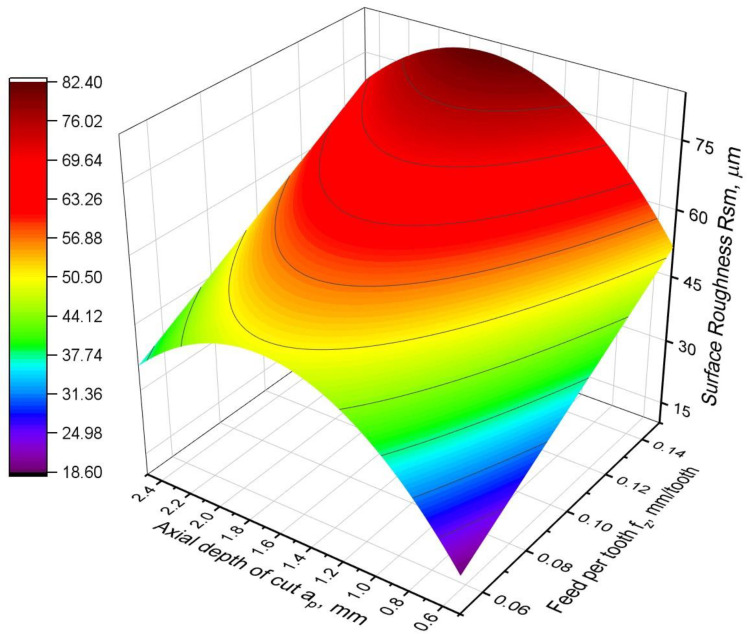
Surface roughness RSm parameter model.

**Table 1 materials-16-02829-t001:** Chemical composition of 42CrMo4 steel (%).

C	Mn	Si	P	S	Cr	Mo
0.38–0.45	0.6–0.9	max 0.4	max 0.025	max 0.035	0.9–1.2	0.15–0.3

**Table 2 materials-16-02829-t002:** Experimental test plan.

Test Number	Feed per Tooth *f_z_*, mm/Tooth	Axial Infeed *a_p_*, mm
1	0.10	1.5
2	0.15	0.5
3	0.05	1.5
4	0.10	2.5
5	0.10	1.5
6	0.05	0.5
7	0.10	1.5
8	0.15	2.5
9	0.05	2.5
10	0.10	1.5
11	0.10	1.5
12	0.10	0.5
13	0.15	1.5

**Table 3 materials-16-02829-t003:** Cutting force values.

Test Number	F_fN_, N (Real)	F_fN_, N (FEM)	F_fN_, % (Dev)	F_f_, N (Real)	F_f_, N (FEM)	F_f_, % (Dev)	F_a_, N (Real)	F_a_, N (FEM)	F_a_, % (Dev)
1	528	570	−8.0	193	239	−23.8	258	244	5.4
2	289	317	−9.7	177	170	4.0	152	153	−0.7
3	372	421	−13.2	155	170	−9.7	221	235	−6.3
4	812	895	−10.2	305	337	−10.5	331	300	9.4
5	515	570	−10.7	196	239	−21.9	249	244	2.0
6	172	190	−10.5	89	91	−1.9	144	138	4.2
7	532	570	−7.1	192	239	−24.5	256	244	4.7
8	1033	1087	−5.2	415	442	−6.5	377	357	5.3
9	588	628	−6.8	206	225	−9.2	321	303	5.6
10	519	570	−9.8	199	239	−20.1	252	244	3.2
11	533	570	−6.9	202	239	−18.3	268	244	9.0
12	288	320	−11.1	121	135	−11.6	165	158	4.2
13	645	661	−2.5	302	312	−3.3	278	266	4.3

**Table 4 materials-16-02829-t004:** Analysis of variance for the F_fN_ cutting force component model.

Term	DF	Seq SS	C (%)	Adj SS	Adj MS	F-Value	*p*-Value
Model	3	615,743	99.48%	615,743	205,248	577.24	0.000
Linear	2	588,847	95.14%	588,847	294,423	828.04	0.000
*f_z_*	1	116,204	18.77%	116,204	116,204	326.82	0.000
*a_p_*	1	472,643	76.36%	472,643	472,643	1329.27	0.000
*f_z_∙a_p_*	1	26,896	4.35%	26,896	26,896	75.64	0.000
Error	9	3200	0.52%	3200	356		
Lack of Fit	5	2943	0.48%	2943	589	9.15	0.026
Pure Error	4	257	0.04%	257	64		
Total	12	618,943	100.00%				

**Table 5 materials-16-02829-t005:** Analysis of variance for the F_f_ cutting force component model.

Term	DF	Seq SS	C (%)	Adj SS	Adj MS	F-Value	*p*-Value
Model	4	86,546.8	99.38%	86,546.8	21,636.7	322.00	0.000
Linear	2	81,171.3	93.21%	81,171.3	40,585.7	604.00	0.000
*f_z_*	1	32,808.7	37.67%	32,808.7	32,808.7	488.26	0.000
*a_p_*	1	48,362.7	55.54%	48,362.7	48,362.7	719.74	0.000
Square	1	1695.8	1.95%	1695.8	1695.8	25.24	0.001
*f_z_∙f_z_*	1	1695.8	1.95%	1695.8	1695.8	25.24	0.001
Two-Way	1	3679.6	4.23%	3679.6	3679.6	54.76	0.000
*f_z_∙a_p_*	1	3679.6	4.23%	3679.6	3679.6	54.76	0.000
Error	8	537.6	0.62%	537.6	67.2		
Lack of Fit	4	468.4	0.54%	468.4	117.1	6.77	0.045
Pure Error	4	69.2	0.08%	69.2	17.3		
Total	12	87,084.3	100.00%				

**Table 6 materials-16-02829-t006:** Analysis of variance for the F_a_ cutting force component model.

Term	DF	Seq SS	C (%)	Adj SS	Adj MS	F-Value	*p*-Value
Model	2	56,210.8	97.23%	56,210.8	28,105.4	175.45	0.000
Linear	2	56,210.8	97.23%	56,210.8	28,105.4	175.45	0.000
*f_z_*	1	2440.2	4.22%	2440.2	2440.2	15.23	0.003
*a_p_*	1	53,770.7	93.01%	53,770.7	53,770.7	335.66	0.000
Error	10	1601.9	2.77%	1601.9	160.2		
Lack of Fit	6	1390.7	2.41%	1390.7	231.8	4.39	0.087
Pure Error	4	211.2	0.37%	211.2	52.8		
Total	12	57,812.8	100.00%				

**Table 7 materials-16-02829-t007:** Analysis of variance for the Ra surface roughness model.

Term	DF	Seq SS	C (%)	Adj SS	Adj MS	F-Value	*p*-Value
Model	4	0.011087	94.77	0.011087	0.002772	36.21	0.000
Linear	2	0.009525	81.42	0.009525	0.004763	62.23	0.000
*f_z_*	1	0.008042	68.74	0.008042	0.008042	105.08	0.000
*a_p_*	1	0.001483	12.68	0.001483	0.001483	19.38	0.002
Square	2	0.001562	13.35	0.001562	0.000781	10.20	0.006
*f_z_∙f_z_*	1	0.000122	1.04	0.000608	0.000608	7.94	0.023
*a_p_∙a_p_*	1	0.001440	12.31	0.001440	0.001440	18.81	0.002
Error	8	0.000612	5.23	0.000612	0.000077		
Lack of Fit	4	0.000492	4.21	0.000492	0.000123	4.10	0.100
Pure Error	4	0.000120	1.03	0.000120	0.000030		
Total	12	0.011699	100.00				

**Table 8 materials-16-02829-t008:** Analysis of variance for the Rz surface roughness model.

Term	DF	Seq SS	C (%)	Adj SS	Adj MS	F-Value	*p*-Value
Model	3	1.78109	84.26%	1.78109	0.593698	16.06	0.001
Linear	2	0.96363	45.59%	0.96363	0.481815	13.03	0.002
*f_z_*	1	0.91260	43.17%	0.91260	0.912600	24.68	0.001
*a_p_*	1	0.05103	2.41%	0.05103	0.051030	1.38	0.270
Square	1	0.81747	38.67%	0.81747	0.817465	22.11	0.001
*a_p_∙a_p_*	1	0.81747	38.67%	0.81747	0.817465	22.11	0.001
Error	9	0.33274	15.74%	0.33274	0.036971		
Lack of Fit	5	0.32000	15.14%	0.32000	0.063999	20.09	0.006
Pure Error	4	0.01274	0.60%	0.01274	0.003186		
Total	12	2.11383	100.00%				

**Table 9 materials-16-02829-t009:** Analysis of variance for the RSm surface roughness model.

Term	DF	Seq SS	C (%)	Adj SS	Adj MS	F-Value	*p*-Value
Model	3	3531.71	95.82%	3531.71	1177.24	68.71	0.000
Linear	2	2061.35	55.93%	2061.35	1030.68	60.15	0.000
*f_z_*	1	1633.50	44.32%	1633.50	1633.50	95.34	0.000
*a_p_*	1	427.85	11.61%	427.85	427.85	24.97	0.001
Square	1	1470.36	39.89%	1470.36	1470.36	85.82	0.000
*a_p_∙a_p_*	1	1470.36	39.89%	1470.36	1470.36	85.82	0.000
Error	9	154.20	4.18%	154.20	17.13		
Lack of Fit	5	121.00	3.28%	121.00	24.20	2.92	0.161
Pure Error	4	33.20	0.90%	33.20	8.30		
Total	12	3685.91	100.00%				

## Data Availability

Data available on request.

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
