# Peer review of "Modelling of the Face-Milling Process by Toroidal Cutter"

_materials, 2023, doi:10.3390/ma16072829_

Round 1

Reviewer 1 Report

This paper deals with the study of experimental and simulation of the face milling process of a flat surface with toroidal milling cutter. The numerical part and the comparison results between experiments and simulations are good. The roughness observations can lead to optimize cutting conditions. Nevertheless, this work presents a lack of structure, which could be improved the attention of the reader. Some comments were done as followed in order to improve the quality of the paper:

1) The first sentence of the abstract is useless. The abstract must be precise and concise and do not exceed 200 words.

2) The introduction is good, but what is the usefulness of Fig 1, particularly it is not referred in the text?

Furthermore, Authors must present methodology and the plan of the article at the very end of the introduction.

3) Part 2 should be entitled “2. Experimental and numerical procedure”. Authors can provide “2.1 Numerical background” and “2.2 Experiment procedure” (right after Fig.4).

4) In section 2, have some friction tests been done to implement the Coulomb friction model?

Moreover, it is hard to understand the machining context. It could be interesting to know the cutting parameters and why such parameters were used in this study.

5) Regarding to the cutting forces, in section 3, are the interpreted forces from Kistler table or from the carbide insert reference? Authors may add a Fig to illustrate these cutting forces.

6) How was the roughness measured. Authors must define all experiments used in this study in order to do the same experiments by others. Regarding to surface roughness, why measured 3 roughness criteria?

7) Fig 10 and 11 must have the scale on the illustrations.

8) Machining with emulsion has been studied (comparison between Fig 10 and 11) but has not been defined at the beginning of the paper. It is inadequate. Authors have to make an effort to describe what they are going to do and why, either at the end of the introduction or at the beginning of the section 2.

9) Conclusion must be improved. There are too much bullet points. Some point could be combined. The conclusion must be opened by announcing the prospects at the end.

Author Response

Thank you very much for your comments. We have analysed the article carefully and considered all suggestions. Any changes made to the article have been noted in red. We are also now providing answers to the questions and comments:

1) The first sentence of the abstract is useless. The abstract must be precise and concise and do not exceed 200 words.

The first sentence of the summary has been removed. After the change, the summary has 198 words.

2) The introduction is good, but what is the usefulness of Fig 1, particularly it is not referred in the text? Furthermore, Authors must present methodology and the plan of the article at the very end of the introduction.

Figure 1 explains how the depth of cut affects the cross-sectional area of the cut layer and the enering angle. This is important because the cross-sectional area of the cut layer has a key influence on the cutting process. A reference to Figure 1 has been added in the text. As suggested by the reviewer, the methodology and plan of the article have been added at the end of Chapter 1.

3) Part 2 should be entitled “2. Experimental and numerical procedure”. Authors can provide “2.1 Numerical background” and “2.2 Experiment procedure” (right after Fig.4).

A suggested change to the title of Part 2 has been made and the chapter has been divided into subsections.

4) In section 2, have some friction tests been done to implement the Coulomb friction model?

Moreover, it is hard to understand the machining context. It could be interesting to know the cutting parameters and why such parameters were used in this study.

Friction tests have not been carried out. The Coulomb friction model is the internal model used in the AdvantEdge system. Based on literature data (Grzesik, W.; Nieslony, P. Prediction of Friction and Heat Flow in Machining Incorporating Thermophysical Properties of the Coating–Chip Interface. Wear 2004, 256), the coefficient of friction for coating tool-steel pairs was set at 0.4. The cutting parameters used in the experiment are shown in Table 2. The range of cutting parameters was set based on the tool manufacturer's recommendations so that the tool cuts within the range of optimal cutting parameters.

5) Regarding to the cutting forces, in section 3, are the interpreted forces from Kistler table or from the carbide insert reference? Authors may add a Fig to illustrate these cutting forces.

Figure 5 was corrected and the system of cutting force components was added. The actual cutting forces recorded with the dynamometer were interpreted in the dynamometer system as shown in Figure 5.

6) How was the roughness measured. Authors must define all experiments used in this study in order to do the same experiments by others. Regarding to surface roughness, why measured 3 roughness criteria?

Roughness measurements were carried out using a profilometer. The surface roughness was measured at 5 locations and then the average value of the roughness was determined. A dispersion of less than 10 per cent was achieved in all measurements. Measurements were taken along the feed direction at the centre of the cutting width. The elementary length was equal to 0.8 mm and the measurement distance was equal to 5.6 in accordance with EN ISO 21920:2022. 3 roughness parameters were measured, as each provided different information about the surface profile and the parameters are complementary. In addition, Ra and Rz are the most commonly used in practice and have the greatest utilitarian value. Ra averages the profile, Rz responds to random roughness and RSm provides information on the horizontal distribution of roughness. Information on surface roughness measurement conditions has been added to the article.

7) Fig 10 and 11 must have the scale on the illustrations.

Scale has been added to Figures 10 and 11.

8) Machining with emulsion has been studied (comparison between Fig 10 and 11) but has not been defined at the beginning of the paper. It is inadequate. Authors have to make an effort to describe what they are going to do and why, either at the end of the introduction or at the beginning of the section 2.

Information on machining conditions has been added. Data on the coolant used is provided. In addition, the methodologies and plan of the article are included at the end of introduction.

9) Conclusion must be improved. There are too much bullet points. Some point could be combined. The conclusion must be opened by announcing the prospects at the end.

The conclusion of the article has been improved. Particular attention has been paid to improving the conclusions. In addition, perspectives for further research have been added.

Reviewer 2 Report

Gdula is too much mentioned, please reduce, because there are a bunch of people missed in the state of the art, G.Urbikain, working with barrel-shape tools, R. Polvorosa, O.Pereira, and others, when the face milling and other end mills were used. Please rewrite the introduction.

Conclusions must be reduced to those really meaningful. The half of them are common knowledge. ANOVA must all time be used in comparison with the state of the art. Please improve the discussion about results.

Amigo et al. worked on process in which wear is key, that would be the case in your face milling, see Measurement, 112580 High-feed machining strategies were launched by tool developers. The main idea is ​​using extremely low side cutting edge angles Kr so that high feeds and low chip thicknesses are both possible. 

Round inserts are very common, so the main paper ideas can be only worth if a real state of the art is offered to readers.

Figures 10 and 11…scales?

Figure 5 does not show anything: can you give a zoom and use labels for the details.

The cutting forces were first determined in AdvantEdge using the FEM: this is only allowed when a deep discussion is possible, lamikiz, salgado en others gave values for real forces. Use the real forces by kistler, the FEM ones are not real. What was the coolant approach?

Niesony, Grzesik and Habrat: this is Niesony et al. please check the style of references.

Figure 1 shows the equivalent side or lead angle when round insert are used, depending on the tool engagement. The idea was suggested by F.j. campa when thin floors and thin walls are milled. No any mention of his7her work.

[1–8]. This is simply unacceptable. Please define which is the main idea of each work.

Reduce length of the paper, more details in figures

Next version must be very improved: introduction, list of works, figures…

Author Response

Thank you very much for your comments. We have analysed the article carefully and considered all suggestions. Any changes made to the article have been noted in red. We are also now providing answers to the questions and comments:

Gdula is too much mentioned, please reduce, because there are a bunch of people missed in the state of the art, G.Urbikain, working with barrel-shape tools, R. Polvorosa, O.Pereira, and others, when the face milling and other end mills were used. Please rewrite the introduction.

The introduction has been corrected in accordance with the reviewer's comment. The literature list has been revised and supplemented with papers on cutting with round insert tools.

Conclusions must be reduced to those really meaningful. The half of them are common knowledge. ANOVA must all time be used in comparison with the state of the art. Please improve the discussion about results.

The conclusions have been corrected. A discussion of the research findings has also been completed.

Amigo et al. worked on process in which wear is key, that would be the case in your face milling, see Measurement, 112580 High-feed machining strategies were launched by tool developers. The main idea is using extremely low side cutting edge angles Kr so that high feeds and low chip thicknesses are both possible. 

Amigo's work was taken into account in the state of the art analysis.

Round inserts are very common, so the main paper ideas can be only worth if a real state of the art is offered to readers.

The reviewer's comment has been applied and articles on the study of the milling process with round insert tools have been added to the literature review.

Figures 10 and 11…scales?

Scale has been added to Figures 10 and 11.

Figure 5 does not show anything: can you give a zoom and use labels for the details.

Figure 5 has been corrected. More details of the test stand have been added. The components of the test stand have been described. The system of cutting force components that were measured during testing is presented.

The cutting forces were first determined in AdvantEdge using the FEM: this is only allowed when a deep discussion is possible, lamikiz, salgado en others gave values for real forces. Use the real forces by kistler, the FEM ones are not real. What was the coolant approach?

Firstly, the cutting forces were determined using the AdvantEdge software. The purpose of simulating the cutting process using the FEM was to verify the validity of the range of cutting parameters tested. In addition, comparisons with experimental results were made to check whether the power law model implemented in AdvantEdge performs well in the analysis of face milling with a toroidal milling cutter. As evidenced by the results from the publication (Habrat, W.; Nieslony, P.; Grzesik, W. Experimental and Simulation Investigations of Face Milling Process of Ti-6Al-4V Titanium Alloy. Advances in Manufacturing Science and Technology 2015, 39,) different models (PL, JC) can give different results, sometimes significantly deviating from reality. In the following part of the article, the actual cutting forces were analysed. Cooling conditions are included in the article.

Niesony, Grzesik and Habrat: this is Niesony et al. please check the style of references.

The reference style was checked. The reference style for the Materials journal in Zotero was applied.

Figure 1 shows the equivalent side or lead angle when round insert are used, depending on the tool engagement. The idea was suggested by F.j. campa when thin floors and thin walls are milled. No any mention of his7her work.

A reference was made to Camp's article in the introduction.

[1–8]. This is simply unacceptable. Please define which is the main idea of each work.

The reviewer's comment has been complied with. Literature references have been corrected.

Reduce length of the paper, more details in figures

Next version must be very improved: introduction, list of works, figures…

All reviewer comments have been applied. Figures have been corrected and completed. All the reviewer's suggested articles were included in the introduction. Taking into account the comments of all reviewers, we unfortunately had to complete the article and were not able to shorten it.

Reviewer 3 Report

Please, see the file attached with my suggestions.

Author Response

Thank you very much for your comments. We have analysed the article carefully and considered all suggestions. Any changes made to the article have been noted in red. We are also now providing answers to the questions and comments:

 thank the journal for trust me as reviewer. Let me kindly give you some advice. The last history of milling cannot be understood without the evolution of Prof Altintas, in the work of Lopez de Lacalle, and some others who did a great step beyond modelling. I see you did not include any work from the school, and that is not a logical fact in 2023. Modelling is like a tree, after the trunk by Altintas, the main branch in curved edge milling cutter were others.

Suggested literature items in the introduction are included.

− In toroidal milling, a cutting tool with a toroidal shape (i.e., a tool with a rounded or circular shape) is used to cut a groove or cavity into a workpiece. The toroidal shape of the tool allows it to create a curved or rounded shape in the workpiece, which can be useful for creating complex shapes or contours.This type of milling is often used in the production of molds and dies, as well as in the manufacture of aerospace components, medical devices, and other precision parts. The process can be performed on a variety of materials, including metals, plastics, and composites.

The subject of the study was the process of horizontal milling of flat surfaces. This was due to the fact that very often toroidal cutters are used for face milling of flat surfaces instead of cutters with square cutting inserts.

− Toroidal milling is a specialized machining technique that can be used to create intricate shapes and contours in a wide range of materials, making it an important tool in many manufacturing industries.The ideas by Lamikiz or Lopez de Lacalle were followed by G.Urbikain, in basic modelling terms. On the other hand when coolants were introduced the works of O.Pereira or R.Polvorosa, or Wretland are to be noticed. My humble guessing is that you could not access the main work by the above leading authors, but that van be solve in next version.

Suggested literature items in the introduction are included.

− Figure 1 is from a handbook or is it yours? Did you mention dynamic problems or vibrations please check Urbikain theory, in J of Sound and vibration and others.

Figure 1 is of our authorship. A reference to this figure has been added. The problems of the dynamics of the face milling process are very important. Thank you for the suggestion. The problem of stability of the face milling process was presented in the introduction. However, the stability of the machining process was not analysed in this article.

− Coulomb frictional mode is not accurate, please discuss it better. I do not see that FEM help you a lot here.

The AdvantEdge software was used in the FEM analysis. The article presents the mathematical models that this software uses to perform the calculations. Among other things, a Coulomb friction model is implemented in AdvantEdge. This model was not analysed. Information on the assumed friction coefficient was added in the article. FEM analyses were performed for two reasons. Firstly, the FEM analyses provided information on the expected cutting forces and allowed verification of the correctness of the range of parameters tested. Secondly, comparison of the FEM results with the experimental data made it possible to verify the suitability and accuracy of the Power Law model implemented in AdvantEdge for modelling the face milling process with a toroidal cutter. As shown by the results from the publication (Habrat, W.; Nieslony, P.; Grzesik, W. Experimental and Simulation Investigations of Face Milling Process of Ti-6Al-4V Titanium Alloy. Advances in Manufacturing Science and Technology 2015, 39,) different models (PL, JC) can give different results, sometimes significantly deviating from reality.

− 42CrMo4 steel with a hardness of 220 167 HB: are you sure, is it not heat-treated? I see in table 1 the composition but I do not see the chemical analyses. Is it a standard table?

The steel tested was in the softened state hence its hardness was 220 HB. It had not been heat treated. Table 1 shows the chemical composition of the steel. No additional chemical analyses were performed.

− Sandvik tool is a commercial one. Please provide more information. The angles are very important, the grade, the coating…in present definition all is missed.

As suggested, the information on the cutting insert used has been supplemented in the article.

− The statistical analysis is very weak, why did you not replay several tests? Test repetition refers to the act of repeating a test or experiment multiple times to ensure the reliability and validity of the results obtained.

An experimental plan was generated according to DOE rules. A Central Composite Fractional Design plan with 3 levels and 2 factors was used. According to the plan, there were two variable parameters. Each parameter had three values, giving 9 variations. There were 13 cutting tests with 4 repetitions. A detailed description of the cutting tests is provided in Table 2.

− Picture scales, no any dimensions there.

Scale was added in figures 10 and 11

− ANOVA IS VERY LONG: WHICH ONES ARE YOUR FINDISNG?

The main finding from the Anova analysis is the description of the dependence of roughness parameters on technological parameters. It was shown that the depth of cut has a non-monotonic effect on the roughness parameters and that the maximum roughness occurs for a depth of cut of about 2 mm, which is for a 45-degree entering angle. The analysis of the cutting force components, on the other hand, shows that depth of cut has a monotonic effect on the cutting force components. The greater the depth of cut, the greater the difference between the component normal to the feed and the other components. Furthermore, it was observed that the feed and axial force components increase linearly with increasing depth of cut. This means that changing the entering angle does not affect the relationship between these components.

− How it is possible feed force would be much higher than tangential force? Introduce a scheme of coordinate axes in some pictures.

Figure 5 has been completed and a description of the test stand has been added, as well as showing the system of cutting force components. This system was related to the force gauge, hence the division of the components into axial, feed and normal to feed was done.

− Please go to see Prediction of specific force coefficients from a FEM cutting model, The International Journal of Advanced Manufacturing Technology 43 (3-4), 348-356 Authors are using the idea in similar cases to yours: FEM plus mechanistic approaches. − I am very sorry to inform you that this paper requires significant revisions. I kindly request that you do your utmost best to improve it. Thank you very much for your efforts.

The suggested literature position in the introduction is included.

Next version need a serious introduction. Just on the contrary, more data would be required, surface roughness testing: This involves measuring the roughness of the machined surface using a profilometer. Surface roughness can affect the performance of the component, particularly in terms of friction and wear. Metallographic analysis: This involves examining the microstructure of the machined material using microscopy. Metallographic analysis can reveal important information about the material's grain structure, porosity, and other features that can affect its mechanical properties.

The reviewer's suggestions have been taken into account. The introduction has been corrected and suggested literature items have been included. The analysis of the roughness measurements has been completed. A metallographic analysis of the samples after milling was not carried out mainly due to the fact that the material was machined in a softened state. The final structure of the material is formed during heat treatment, which is usually applied after machining.

− Dimensional inspection: This involves measuring the dimensions of the machined part to ensure that it meets the required specifications. Dimensional inspection can be done

No dimensional inspection of the workpieces was carried out, as this was not the subject of the analyses. Flat surfaces were milled in the study. In this case, tool deflection has a marginal effect on the workpiece dimension. Our research focused mainly on the analysis of the cutting force components and the roughness of flat surfaces. This is because toroidal milling cutters are often used for milling flat surfaces instead of mills with rectangular inserts.

Round 2

Reviewer 1 Report

Authors have made great efforts to improve the quality of the paper. I suggest to accept this work in this form for publication.

Reviewer 2 Report

GOOD WORK

Reviewer 3 Report

My comments were taken into account. Accepted.